# Targeting Amino Acid Metabolic Reprogramming via L-Type Amino Acid Transporter 1 (LAT1) for Endocrine-Resistant Breast Cancer

**DOI:** 10.3390/cancers13174375

**Published:** 2021-08-30

**Authors:** Haruhiko Shindo, Narumi Harada-Shoji, Akiko Ebata, Miku Sato, Tomoyoshi Soga, Minoru Miyashita, Hiroshi Tada, Masaaki Kawai, Shinkichi Kosaka, Koji Onuki, Shin Usami, Shozo Furumoto, Shinichi Hayashi, Takaaki Abe, Takashi Suzuki, Takanori Ishida, Hironobu Sasano

**Affiliations:** 1Department of Breast and Endocrine Surgical Oncology, Tohoku University Graduate School of Medicine, 2-1 Seiryo-machi, Aoba-ku, Sendai 980-8575, Japan; haruhiko.shindo.s4@dc.tohoku.ac.jp (H.S.); akiko.ebata@med.tohoku.ac.jp (A.E.); miku.s@surg.med.tohoku.ac.jp (M.S.); atihsayim8m8@med.tohoku.ac.jp (M.M.); hiroshi-tada@med.tohoku.ac.jp (H.T.); shinkichi.kosaka.b6@tohoku.ac.jp (S.K.); takanori@med.tohoku.ac.jp (T.I.); 2Department of Pathology Tohoku University Hospital, 1-1 Seiryo-machi, Aoba-Ku, Sendai 980-8574, Japan; hsasano@patholo2.med.tohoku.ac.jp; 3Institute for Advanced Biosciences, Keio University, Tsuruoka, Yamagata 997-0052, Japan; soga@sfc.keio.ac.jp; 4Department of Surgery I, Yamagata University Graduate School of Medical Science, Yamagata 990-9585, Japan; masaaki-kawai@med.id.yamagata-u.ac.jp; 5Department of Breast Surgery, Miyagi Cancer Center, 47-1 Nodayama, Shiote, Medeshima, Natori 981-1293, Japan; koji-onuki@miyagi-pho.jp; 6Department of Breast and Endocrine Surgery, Iwate Prefectural Central Hospital, 1-4-1 Ueda, Morioka 020-0066, Japan; shin-u@fg7.so-net.ne.jp; 7Cyclotron and Radioisotope Center, Tohoku University, 6-3 Aramaki-aza-Aoba, Aoba-ku, Sendai 980-8578, Japan; shozo.furumoto.b6@tohoku.ac.jp; 8Department of Molecular and Functional Dynamics, Tohoku University Graduate School of Medicine, 2-1 Seiryo-machi, Aoba-Ku, Sendai 980-8575, Japan; shin@med.tohoku.ac.jp; 9Division of Medical Science, Tohoku University Graduate School of Biomedical Engineering and Department of Clinical Biology and Hormonal Regulation, Tohoku University Graduate School of Medicine, 2-1 Seiryo-machi, Aoba-ku, Sendai 980-8575, Japan; takaabe@med.tohoku.ac.jp; 10Department of Pathology and Histotechnology, Tohoku University Graduate School of Medicine, 2-1 Seiryo-machi, Aoba-ku, Sendai 980-8575, Japan; t-suzuki@patholo2.med.tohoku.ac.jp

**Keywords:** SLC7A5 (LAT1), SLC43A1 (LAT3), breast cancer, hormone therapy, amino acid metabolism, JPH203

## Abstract

**Simple Summary:**

L-type amino acid transporters such as LAT1 and LAT3 are associated with the uptake of essential amino acids. In particular, LAT1 regulates mammalian target of rapamycin complex 1 (mTORC1) signaling and cell proliferation by regulating leucine uptake. The purpose of this study is to clarify amino acid metabolism via LAT1 and LAT3 in breast cancer and the potential roles of LAT1 in the development of therapeutic resistance and clinical outcome of the patients. Results demonstrated that high LAT1 status was associated with tumor progression in breast cancer patients who received neoadjuvant hormone therapy (NAH), and LAT1 expression in the estrogen deprivation-resistant (EDR) breast carcinoma cell lines were upregulated. JPH203, a selective LAT1 inhibitor, demonstrated inhibitory effects on cell proliferation in the EDR cells. Therefore, LAT1 could serve not only as a prognosis biomarker but also a therapeutic target in estrogen receptor (ER)-positive breast cancer patients.

**Abstract:**

The PI3K/Akt/mTOR pathway has been well known to interact with the estrogen receptor (ER)-pathway and to be also frequently upregulated in aromatase inhibitor (AI)-resistant breast cancer patients. Intracellular levels of free amino acids, especially leucine, regulate the mammalian target of rapamycin complex 1 (mTORC1) activation. L-type amino acid transporters such as LAT1 and LAT3 are associated with the uptake of essential amino acids. LAT1 expression could mediate leucine uptake, mTORC1 signaling, and cell proliferation. Therefore, in this study, we explored amino acid metabolism, including LAT1, in breast cancer and clarified the potential roles of LAT1 in the development of therapeutic resistance and the eventual clinical outcome of the patients. We evaluated LAT1 and LAT3 expression before and after neoadjuvant hormone therapy (NAH) and examined LAT1 function and expression in estrogen deprivation-resistant (EDR) breast carcinoma cell lines. Tumors tended to be in advanced stages in the cases whose LAT1 expression was high. LAT1 expression in the EDR cell lines was upregulated. JPH203, a selective LAT1 inhibitor, demonstrated inhibitory effects on cell proliferation in EDR cells. Hormone therapy changed the tumor microenvironment and resulted in metabolic reprogramming through inducing LAT1 expression. LAT1 expression then mediated leucine uptake, enhanced mTORC1 signaling, and eventually resulted in AI resistance. Therefore, LAT1 could be the potential therapeutic target in AI-resistant breast cancer patients.

## 1. Introduction

Hormone therapy against the estrogen receptor (ER) pathway is the most common therapy in ER-positive breast cancer patients. Initial response rates have become markedly improved but it is also true that some patients still demonstrated de novo or acquired endocrine resistance. Therefore, novel and more effective therapeutic strategies are warranted at this juncture [1].

Many cancers have been reported to maintain their demand for cell growth by adopting extensively distinct metabolic processes. In the tumor microenvironment, reprogrammed glucose, amino acid, and lipid metabolism supply unlimited tumor progression. In addition, metabolic reprogramming is also well known to be closely associated with the resistance to chemotherapy, radiotherapy, and immunotherapy, and to result in the adverse clinical outcome of the patients [2]. In metabolic analysis using capillary electrophoresis time-of-flight mass spectrometry (CE-MS) in breast cancer patients, the metabolism of amino acids was reported to be reprogrammed in invasive ductal carcinoma cells [3]. Therefore, we hypothesized that intracellular metabolic changes, particularly those in amino acid metabolism, could be associated with endocrine resistance in ER-positive breast cancer patients.

Essential amino acids (EAAs) imported by several amino acid transporters have been generally considered pivotal for cell proliferation because the depletion of even a single EAA in vitro induced cell death [4,5]. Among those transporters, the L-type amino acid transporter (LAT1) family is essential for EAAs uptake and comprises four members (LAT1-LAT4); LAT1 has been proposed especially fundamental to cancer viability and reported to be abundant in malignant cells compared to normal cells [6,7]. LAT1 was also reported to transport various EAAs, such as leucine, isoleucine, valine, phenylalanine, tyrosine, tryptophan, methionine, and histidine [8]. LAT1 expression was also reported to be correlated with poor clinical outcomes of the patients in several human malignancies, such as gastric and prostate cancers [9].

The LAT1 was also reported to interact with the scaffolding protein LLGL2 to mediate leucine for maintenance of cell proliferation in ER-positive breast carcinoma cells [10]. The overexpression of both LLGL2 and LAT1 in cancer cells has been also regarded as an inducer of therapeutic resistance to hormone therapy [10]. The PI3K/Akt/mTOR pathway, which is known to interact with the ER pathway, was reported to be frequently upregulated in aromatase inhibitor (AI)-resistant breast cancer [11]. In addition, intracellular levels of free amino acids, in particular, leucine, regulated the mammalian target of rapamycin complex 1 (mTORC1) activation and LAT1 status could regulate leucine uptake, mTORC1 signaling, and cell proliferation [12].

LAT3 was also reported to be overexpressed especially in androgen-sensitive prostate cancer. LAT3 transcription was activated by androgen receptor (AR) signaling and then lead to leucine uptake, mTORC1 signaling, and cell proliferation in primary prostate cancer. In addition, decreased androgen signaling and LAT3 expression after sequential hormone ablation therapy was reported to initiate the transcription of LAT1 [12,13]. Therefore, amino acid metabolic reprogramming associated with LAT1 and LAT3 has become one of the potentially attractive therapeutic targets in metabolism-directed cancer therapies. The LAT1 inhibitor known as JPH203 also inhibited the growth of renal-cell-carcinoma-derived cells. JPH203 was therefore proposed to suppress LAT1-mediated essential amino acids and inactivate the mTOR signaling pathway. The first-in-human phase I study for JPH203 (*n* = 17, colon, pancreas, bile duct, esophagus, and breast cancer patients) has been performed (Clinical trial registration: UMIN000016546). There were no cases of grade 4 treatment-related adverse effects or death [14], but the details of the status of LAT 1 and 3 in human breast cancer have remained virtually unknown.

Therefore, in this study, we first attempted to clarify amino acid metabolic reprogramming in breast cancer and evaluate the amino acid transporter expression and amino acid metabolites to explore their pathological and clinical significance.

## 2. Materials and Methods

### 2.1. Patients Cohorts and Sample Selection

We examined two cohorts of participants in this study. The first cohort comprised 187 patients with ER-positive primary breast cancer who underwent surgery between 2005 and 2008 at Tohoku University Hospital, Sendai, Japan for surgical specimens. The second cohort consisted of 84 patients with ER-positive primary breast cancer who received neoadjuvant hormone therapy (NAH) and surgery between 2013 and 2020 at Tohoku University Hospital, between 2013 and 2016 at Iwate Prefectural Central Hospital, and between 2007 and 2018 at Miyagi Cancer Center, Japan, and examined biopsy specimens before hormone therapy and surgical specimens after NAH. We defined pathologic complete response (pCR) as the histopathologically confirmed absence of invasive residual carcinoma cells in breast and lymph nodes following careful and extensive histopathological evaluation and residual noninvasive carcinoma cells were allowed [15].

### 2.2. Immunohistochemistry

Immunohistochemistry was performed as previously reported [5]. We used LAT1 mouse monoclonal antibody (1:100, KE023 Trans Genics, Hyogo, Japan) and LAT3 rabbit polyclonal antibody (1:1000, MBL Life science, Tokyo, Japan). We used the Histofine Kit (Nichirei Bioscience, Tokyo, Japan) to employ the streptavidin-biotin amplification method. We visualized the antigen-antibody complex with 3,3′-diaminobenzidine (DAB) solution (1mM DAB, 50mM Tris-HCl buffer (pH 7.6), and 0.006% H2O2) and counterstained with hematoxylin. Human placenta or kidney tissues were used as positive controls.

### 2.3. Scoring of Immunohistoreactivity

We evaluated immnoreactivity of LAT1 using the H score method with modifications. We subsequently obtained LAT1 expression score [total score: T] based on both the relative immunointensity (intensity score: I) in the carcinoma cell membranes and the staining area (proportion score: P), which represented the percentage of the whole carcinoma area, relative to the total section area. (I) was classified as follows: 0 = no staining; 1 = weak; 2 = moderate; and 3 = strong. (P) was classified as follows: 0 = 0%; 1 = 1–10%; 2 = 11–20%; 3 = 21–30%; 4 = 31–40%; 5 = 41–50%; 6 = 51–60%; 7 = 61–70%; 8 = 71–80%; 9 = 81–90%; and 10 = 91–100%. (T) of each patient was scored by multiplying (I) by (P) [16]. Scoring was independently performed by two of the authors (T.S. and H.S.), blinded to each patient’s characteristics. We tentatively designated two groups based on T score, positive (high) and negative (low) LAT1 expression groups ((T) 0–10 and (T) 11–30, respectively) and positive and negative LAT3 expression groups ((T) 0 and (T) 1–30, respectively).

### 2.4. Cell Lines and Culture

MCF-7-E10 breast carcinoma cells were established from MCF-7 cells stably transfected with ERE-GFR reporter plasmids as reported previously [17]. EDR cells (type 1 (EDR1) and type 2 (EDR2) cells: AI-resistant breast cancer models) were established from MCF-7-E10 cells as previously reported [17,18]. E10 cells were cultured in RPMI-1640 medium (Sigma-Aldrich, St Louis, MO, USA) added 10% fetal bovine serum (FBS; Nichirei Bioscience) and 100 μg/mL penicillin/streptomycin (Invitrogen, Carlsbad, CA, USA). EDR1 and EDR2 were kept in phenol red-free RPMI-1640 medium (Gibco BRL, Grand Island, NY, USA) added 5% dextran-coated charcoal-treated FBS (C-FBS; Sigma-Aldrich) and 100 μg/mL penicillin/streptomycin (Invitrogen). The cells were incubated at 37 °C in 5% CO_2_.

### 2.5. Western Blotting

Western blotting was performed as previously reported [19]. The membrane was incubated with primary antibodies against LAT1 (1:250, Cell Signaling Technology Japan, K.K., Tokyo), LAT3 (1:250, MBL Life science, Tokyo, Japan), and β-actin (1:5000, Proteintech, Tokyo, Japan). The whole western blot figures can be found in the Appendix A.

### 2.6. Cell Proliferation Assay

Cells were seeded in 96-well plates containing amino acids free DMEM/F12 (USBiological, #D9807-10), supplemented with 17.5 mM d-glucose, 2.5 mM Gln, and 5% C-FBS. l-Leucine was added before use at concentrations of 4.5 mM for 48 h [10]. We determined cell proliferation using the WST-8 colorimetric assay (Cell Counting Kit-8; Dojindo Laboratories, Kumamoto, Japan, #343-07623). Concisely, we added WST-8 reagent solution to each well, subsequently incubated the microplates for 2 h at 37 °C. We then used a cell counter (Sysmex CDA-500, Sysmex Corporation, Hyogo, Japan) to measure absorbance at 450 nm [20].

Cells were treated with JPH203 (Selleck Chemicals, Sylvanfield Drive, Houston, TX, USA) for 48 h. We determined cell proliferation in the same manner as above [20].

### 2.7. Metabolic Analysis

Sample preparation was previously reported [5]. Metabolome measurement was performed using capillary electrophoresis time-of-flight mass spectrometry at the Institute for Advanced Biosciences, Keio University (Tsuruoka, Japan) [21]. The acquired concentrations of the metabolites were normalized according to the cell numbers. The software (MasterHands) developed by Keio University was used for raw data processing [21].

### 2.8. Amino Acid Uptake with Radiotracers

Cells (5.0 × 10^5^/well) were seeded into 24-well plates for 48 h. They were subsequently exposed to a glucose-free medium (DMEM without glucose added 0.5% BSA and 0.125 mM sodium ascorbate) with 50 μCi/mL of 18 F-fluoro-ethyl-tyrosine (^18^F-FET) for 1 h. The cells were washed with PBS, and lysing cells were suspended in 0.5 M aqueous sodium hydroxide and 0.5 M hydrochloric acid. The γ-counter (AccuFLEX g7000; Hitachi Aloka Medical, Tokyo, Japan) was used for radioactivity measurement. Total protein concentrations were obtained by a protein assay (Qubit ^®^: Thermo Fisher Scientific).

### 2.9. Statistical Analysis

Statistical analysis was carried out using JMP Pro 15 software (SAS Institute, Inc, Cary, NC, USA). The Kaplan–Meier method and the log-rank test were used to analyze survival differences. Disease-free survival (DFS) was defined as the duration from the surgery to the disease recurrence. Breast-cancer-specific survival (BCSS) was defined as the duration from the surgery to breast-cancer-induced death for each patient. Student’s t-test was used for clinicopathological and cell viability analyses. All *p*-values less than 0.05 were considered statistically significant in this study.

## 3. Results

### 3.1. LAT1 Expression Was Associated with Disease-Free Survival and Breast Cancer-Specific Survival in Breast Cancer

We studied the associations of LAT1 status with DFS and BCSS of the patients. The clinicopathological characteristics of cohort 1 (*n* = 187) were summarized in Table 1.

The Kaplan–Meier plots in Figure 1 demonstrated that DFS in the high LAT1 expression group was shorter than that in the low LAT1 expression group (HR: 3.5, 95% CI: 1.6–7.7, *p* = 0.0011). BCSS in the high LAT1 expression group was also significantly shorter than that in the low LAT1 expression group (HR: 3.9, 95% CI: 1.5–10.6, *p* = 0.0035).

### 3.2. LAT1 Expression Was Altered by Neoadjuvant Hormone Therapy

We performed immunohistochemistry using pathology specimens of the patients at both pre-NAH (Pre) and post-NAH (Post) to further explore whether the changes of LAT1 and LAT3 levels were associated with clinicopathological factors. The clinicopathological characteristics of cohort 2 (*n* = 84) were summarized in Table 2. Representative images of LAT1 and LAT3 immunohistochemistry were illustrated in Figure 2. Among 84 patients who received NAH, 36 (42.9%) were tentatively classified as Pre LAT1 positive and 56 (66.7%) were pre-LAT3 -positive.

The correlations between the change in LAT1 and LAT3 in carcinoma cells and clinicopathological features are demonstrated in Table 3. The mean pre-LAT1 score was 10.0 (range, 0–30), the mean post-LAT1 score was 12.5, the mean pre-LAT3 score was 6.5, and the mean post-LAT3 score was 8.2. A high post-LAT1 expression was significantly correlated with the disease stage (*p* = 0.0003), pathological T stage (*p* = 0.0096), pathological N stage (*p* = 0.0365) and histological grade (*p* = 0.0058) of the patients. Conversely, a low post-LAT3 expression was significantly correlated with the disease stage (*p* = 0.0474) and the pathological T stage (*p* = 0.0386) of the patients. Pre-LAT1 and pre-LAT3 were not significantly associated with any of the clinicopathological factors in this study.

We further classified the patients into the following three groups based on the preoperative endocrine prognostic index (PEPI score) as follows: low risk, total score 0; middle risk, total score 1–3; and high risk, Total score 4–12. The correlations between the change in LAT1 and LAT3 status and the co-pathological variables of PEPI score were summarized in Figure 2. As demonstrated in Figure 3, a high PEPI score group tended to enhance LAT1 expression and rather downgrade LAT3 expression although the correlation did not reach statistical significance.

### 3.3. LAT1 Was Overexpressed in Estrogen Deprivation-Resistant Cells (Aromatase Inhibitor-Resistant Breast Carcinoma Model Cells)

The association between LAT1 expression and poor prognosis of the breast cancer patients above did suggest that LAT1 could possibly become a molecular target in breast cancer therapy. We, therefore, aimed to evaluate whether LAT1 inhibition could suppress carcinoma cell proliferation.

We first evaluated the function and expression of LAT1 in EDR cells. E10 cells transfected with ERE-GFR reporter plasmids were used as a comparison in this analysis. Results of western blotting demonstrated that LAT1 was detected in both EDR1 and EDR2 cell lines, with comparable levels (Figure 4a). Whereas, LAT3 was expressed in both E10 and EDR1 cell lines (Figure 4b).

### 3.4. Amino Acids via LAT1 Are Enhanced in Breast Carcinoma Cells

We evaluated ^18^F-FET uptake to further explore the changes in amino acid metabolism in breast carcinoma cells. FET is a tyrosine derivative that could reflect the enhancement of amino acid uptake transported by LAT1 [22]. ^18^F-FET uptake was significantly upregulated in EDR1 and EDR2 cells (FET: *p* = 0.0137 and *p* = 0.0203, respectively; Figure 4c). The analysis of metabolites in these carcinoma cells demonstrated that the concentrations of leucine, isoleucine, and valine, which were branched-chain amino acids (BCAAs) and transported by LAT1, were markedly increased in EDR1 and EDR2 cells. In addition, these two cells did gain TCA intermediates (Figure 4d).

### 3.5. Leucine Uptake Reliefs Nutrient Stress in Estrogen Deprivation-Resistant Cells

We studied whether supplementation with leucine could rescue growth under nutrient stress. Some amino acid transporters are known as antiporters for Gln to facilitate the import of EAAs, and Gln was therefore included in all media. Leucine rescued the proliferation of EDR1 and EDR2 under low glucose conditions, which suggested that leucine uptake via LAT1 supported cell proliferation (Figure 5a).

### 3.6. JPH203 Inhibits the Proliferation of Estrogen Deprivation-Resistant Cells

Functional LAT1 expression and its activity were validated in breast carcinoma cells as above. Therefore, we subsequently explored the effects of JPH203 on cell proliferation.

When the cells were exposed to JPH203, EDR cell viability was significantly decreased in a dose-dependent fashion (Figure 5b). JPH203, therefore, yielded comparable cell growth inhibition in two different types of EDR cells but did not induce any cell death in E10 cells.

These results above indicated that JPH203 exerted inhibitory effects on cell proliferation in breast carcinoma cells.

## 4. Discussion

Despite recent advances in screening, diagnosis, and treatment, drug resistance in breast cancer patients remains an enormous challenge for clinical oncologists. Altered cellular metabolism is one of the hallmarks of malignancy and identifying and targeting specific tumor-associated metabolic signatures could therefore suppress tumor growth and provide more effective therapeutic options even after the development of endocrine resistance in breast cancer patients. In this study, we demonstrated that hormone therapy induced LAT1 expression in ER-positive breast carcinoma cells resulting in tumor progression and that the inhibition of LAT1 function prevented cell proliferation in EDR cells. These findings above did encourage potential administration of LAT1 for breast cancer diagnosis and treatment, although it awaits further investigations.

We first evaluated the LAT1 baseline expression and changes induced by hormone therapy as the high expression of amino acid transporters has been reported in various cancers [23]. Results of our present study did demonstrate that LAT1 expression could indicate amino acid metabolic reprogramming in breast cancer patients as a result of its interaction with DFS and BCSS (Figure 1a,b). In addition, LAT1 expression was detected in carcinoma cells but not in adjacent normal or non-tumorous mammary ductal or epithelial cells (Figure 2d), consistent with the results of the previously reported study [7]. LAT1 was also reported to be significant correlated with the size of the tumor, nuclear grade, and pathological stage in breast cancer patients [24]. Therefore, the results of our present study as well as those of previously reported one did indicate that LAT1 status following hormone therapy was closely correlated with the proliferation in breast cancer. In addition, the results of our present study also demonstrated that LAT1 could predict not only the clinical course of breast cancer patients but also the therapeutic effects of pre/post-operative hormone therapy as demonstrated in Figure 3. Of particular interest, LAT3 expression was demonstrated to be associated with a good prognosis in our present study. The function of LAT3 as an amino acid transporter might be less than that of LAT1 in breast carcinoma cells. However, an evaluation of the clinical significance of LAT3 expression is required for further clarification.

ER signaling was reported to promote LAT1 expression under the lack of nutrition in ER-positive breast carcinoma cell lines [10]. In addition, ER-positive breast cancer patients receiving hormone therapy demonstrated the correlation of high LAT1 expression with unfavorable clinical outcomes [10,25]. Therefore, ER in breast cancer could be involved in LAT1 regulation, as reported in AR of prostate cancer [6]. In our present study, the concentrations of amino acids such as leucine, isoleucine, and valine, imported by LAT1, were significantly higher in EDR cells. These amino acids were all BCAAs, which were essential for protein synthesis and energized cancer cells [26]. Therefore, LAT1 could enhance amino acid uptake required for energy production by activating the TCA cycle using these amino acids. In addition, tumor cells tended to uptake BCAAs [26,27], although its mechanisms have remained unknown. JPH203 has been considered to block LAT1-mediated EAAs uptake by not only decreasing the protein synthesis but also degrading the mTOR signaling pathway [28,29,30,31,32,33], as leucine transported via LAT1 contributed to tamoxifen resistance in ER-positive breast cancer [10].

LAT1 was overexpressed in breast carcinoma cells as demonstrated in our present study. Hence, LAT1-targeted therapy could be feasible in breast cancer patients. The first clinical trials of JPH203 were reported to yield long-term survival for more than 2 years in bile duct cancer patients with the incidence of grade 3 or higher side effects of only 12% in these patients [14]. Dose reduction or delay as a result of side effects could often occur during the course of breast cancer treatment and therefore it has become pivotal to confirm the safety of JPH203. In addition, we also found that a tyrosine derivative tracer; ^18^F-FET uptake was significantly upregulated in EDR cells, which represents that amino-acid-imaging PET could serve as companion diagnostics for JPH203.

A major limitation of this study is the fact that we only studied ER-positive breast cancer patients. Therefore, further studies involving ER-negative breast cancer patients as controls in clinicopathological analyses are required. However, we did demonstrate that hormone therapy-induced LAT1 expression in ER-positive breast cancer cells and that there was a distinctively significant correlation between higher LAT1 expression and poorer prognosis in ER-positive breast cancer patients. In addition, we also demonstrated that functional LAT1 inhibition using JPH203 reduced cell proliferation in breast carcinoma cells. Therefore, LAT1 status is considered to harbor enormous potential as not only a predictor of prognosis or clinical outcome but also a therapeutic target in breast cancer patients.

## 5. Conclusions

In summary, we firstly revealed the metabolic changes in breast cancer patients who received NAH. Hormone therapy changed the tumor microenvironment and caused metabolic reprogramming. Hormone therapy-induced LAT1 expression, which could subsequently mediate leucine uptake and regulate mTORC1 signaling, was correlated with hormone-therapy resistance. Targeting these unique metabolic signatures can potentially constitute another strategy in endocrine-resistant breast cancer patients.

## Figures and Tables

**Figure 1 cancers-13-04375-f001:**
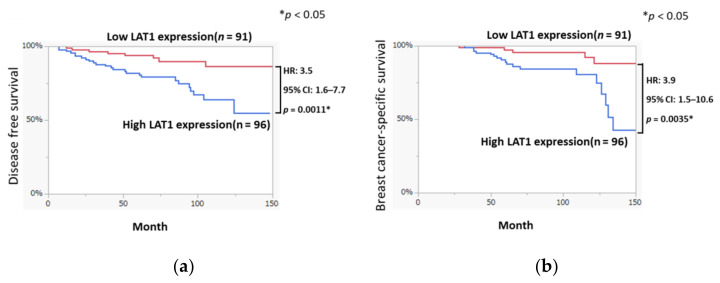
Postoperative survival of ER-positive breast cancer patients classified by LAT1 expression. Cause-specific postoperative survival curves (**a**) disease-free survival [DFS] and (**b**) breast-cancer-specific survival [BCSS] for patients with high and low LAT1 expression levels were demonstrated. The data was analyzed by the Kaplan-Meier method and the log-rank test. * *p* < 0.05.

**Figure 2 cancers-13-04375-f002:**
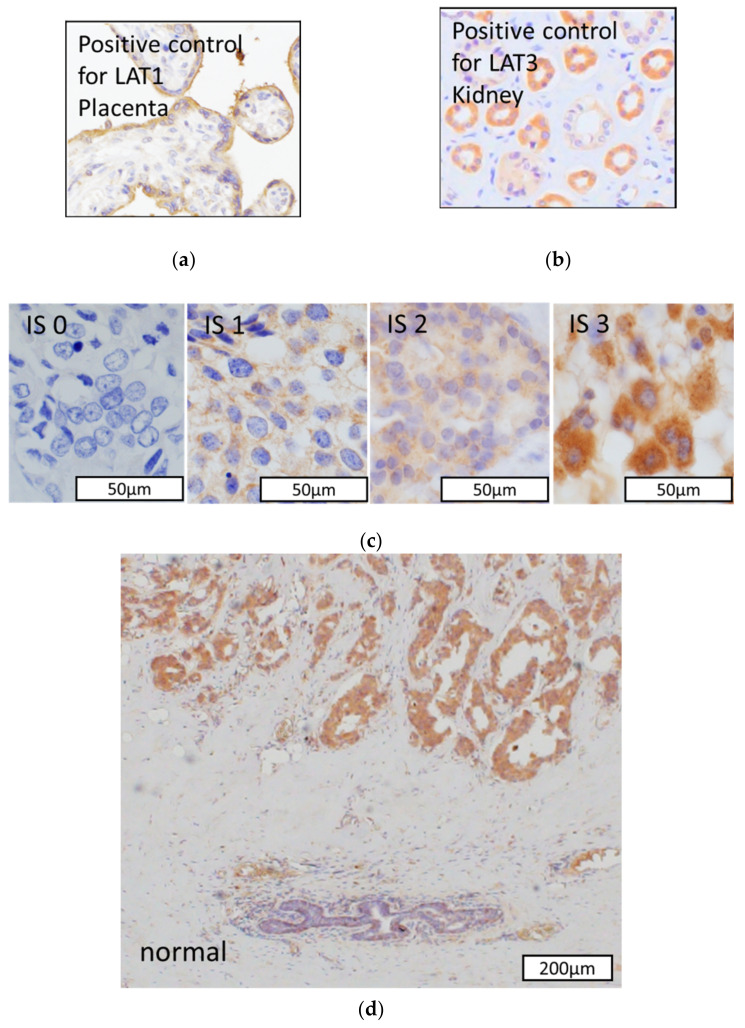
Immunohistochemical analysis of LAT1. Human placenta tissue was used as a positive control (**a**). Immunohistochemistry of LAT3. Human kidney tissue was used as a positive control (**b**). Immunohistochemical staining of LAT1. Relative immunointensity was scored from 0 to 3 (0 = no staining; 1 = week; 2 = moderate; and 3 = strong). Positive immunoreactivity was detected in both cytoplasm and cell membrane (**c**). LAT1 immunoreactivity was abundant in breast carcinoma cells but not in adjacent normal or non-tumorous ductal cells (**d**).

**Figure 3 cancers-13-04375-f003:**
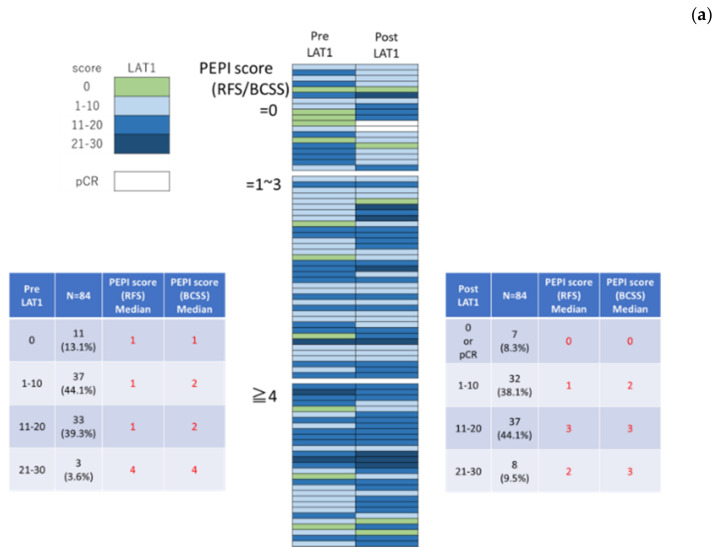
Change of LAT1 (**a**) and LAT3 (**b**) expression tentatively categorized according to PEPI score.

**Figure 4 cancers-13-04375-f004:**
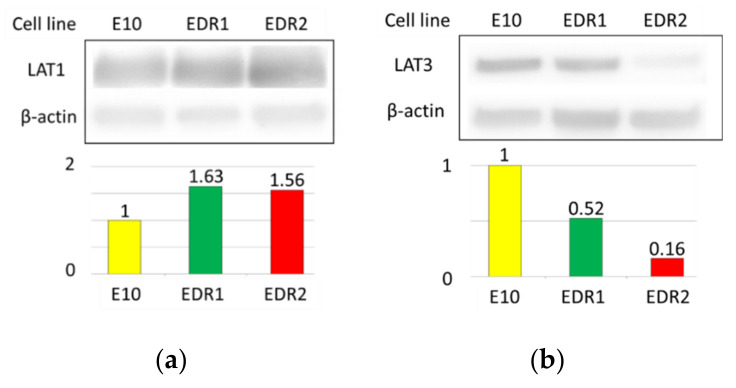
LAT1 (**a**) and LAT3 (**b**) expression in three breast cancer cell lines examined by Western blotting with β-actin as a protein loading control. Cellular uptake of ^18^F-FET radiotracers in three breast carcinoma cell lines (**c**). Metabolome analysis of three breast carcinoma cell lines. These amino acids levels imported by LAT1 corresponded to intermediates of the TCA cycle (**d**). The data was analyzed by the Student’s *t*-test. * *p* < 0.05.

**Figure 5 cancers-13-04375-f005:**
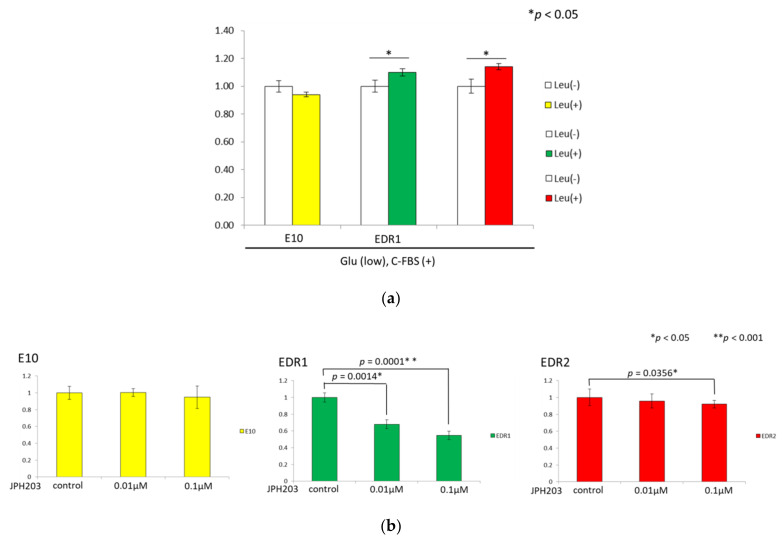
Cell proliferation in three breast cancer cell lines under nutrient stress (**a**). Cell proliferation in three breast cancer cell lines with 0, 0.01, and 0.1 μM JPH203 for 24 h (**b**). The data was analyzed by the Student’s *t*-test. * *p* < 0.05, ** *p* < 0.001.

**Table 1 cancers-13-04375-t001:** Clinicopathological characteristics of ER-positive breast cancer patients (Cohort 1).

	*n* = 187	%		Positive	%
Age, Median (years)	55		LAT1		
	(27–88)		pTis	1	50.0
			pT1	66	48.9
pT			pT2	16	66.7
pTis	2	1.1	pT3	0	0
pT1	135	72.2	pT4	13	59.1
pT2	24	12.8	Total	96	51.3
pT3	4	2.1			
pT4	22	11.8			
pN					
pN0	127	67.9			
pN1<	60	32.1			
Ki-67, Median (%)	10				
	(1–53)				
Premenopausal	67	35.8			
Postmenopausal	120	64.2			

**Table 2 cancers-13-04375-t002:** Clinicopathological characteristics of breast cancer patients with hormone therapy prior to surgery (Cohort 2).

	*n* = 84	%		Positive	
Age, Median (years)	71.5		Pre LAT1		
	(53–90)		cTis	1	50.0
			cT1	12	32.4
cT			cT2	14	63.6
cTis	2	2.4	cT3	1	25.0
cT1	37	44.1	cT4	8	42.1
cT2	22	26.2	Total	36	42.9
cT3	4	4.8			
cT4	19	22.6	Pre LAT3		
			cTis	0	0
cN			cT1	23	62.2
cN0	59	70.2	cT2	14	63.6
cN1<	25	29.8	cT3	4	100
			cT4	15	78.9
Pre Ki-67, Median (%)	13.0		Total	56	66.7
	(0.1–80)				
Medicine of NAH					
ANA	35	41.7			
LET	46	54.8			
EXE	3	3.6			

**Table 3 cancers-13-04375-t003:** Clinicopathological features according to the status of LAT1 and LAT3 (Cohort 2, *n* = 84). The data was analyzed by the Student’s t-test. * *p* < 0.05.

		Pre LAT1		Post LAT1		Pre LAT3		Post LAT3	
		Total Score Mean	*p*-Value	Total Score Mean	*p*-Value	Total Score Mean	*p*-Value	Total Score Mean	*p*-Value
Total		10.0 (0–30)		12.5 (0–30)		6.5 (0–30)		8.2 (0–30)	
pStage 0–I		8.9	0.0778	9.7	0.0003 *	6.6	0.4783	9.3	0.0474 *
II–IV		11.0		15.4		6.5		7.0	
pT is–1		9.6	0.2528	11.1	0.0096 *	6.4	0.3835	9.1	0.0386 *
2–4		10.6		15.1		6.8		6.6	
pN −		9.4	0.1655	11.4	0.0365 *	6.5	0.4981	9.0	0.0519
+		11.0		14.6		6.6		6.7	
Histological Grade	1	9.4	0.2606	10.4	0.0058 *	5.8	0.1661	9.0	0.1254
(HG)	2–3	10.4		14.6		7.1		7.4	
Post Ki-67/ Pre Ki-67	<1.00	9.5	0.1332	12.0	0.1251	6.8	0.2144	8.5	0.258
	1.00≤	11.5		14.3		5.7		7.4	

## Data Availability

The authors confirm that all the relevant data are included in the article and/or its Appendix A Files.

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
