# Peer review of "Targeting Amino Acid Metabolic Reprogramming via L-Type Amino Acid Transporter 1 (LAT1) for Endocrine-Resistant Breast Cancer"

_cancers, 2021, doi:10.3390/cancers13174375_

Round 1

Reviewer 1 Report

The Authors investigated amino acid metabolism and in particular the role of LAT1, in breast cancer patients. Specifically, they explored the potential roles of LAT1 in developing clones resistant to therapy together with its effect on prognosis. In particular, they investigated the clinical setting of primary systemic therapy. Hormonal manipulation was able to change the tumor microenvironment with metabolic reprogramming through induction of LAT1 expression. LAT1 expression was found to mediate for leucine uptake, enhanced mTORC1 signaling, resulting in clonal resitant. These results are meaningful since they may indicate LAT-1 as a potential therapeutic target.

The topic is of interest, the manuscript is clear and well-written, with a reliable methodology, well-defined structure and scientific soundness. I only have one minor comment:

  • I am a bit skeptical about the manuscript title. What is the meaning of stage 1? Why is that?

Reviewer 2 Report

In this manuscript Shindo et al. describe the role of LAT1 in therapeutic resistance to hormonal therapy for estrogen receptor positive breast cancer. 

Study is well done. In the study cohort there were patients who were both pre and post-menopausal. How do authors think LAT1 expression changes with menopause? Also influence on LAT1 to DFS and BCSS influenced by menopausal status, can Figure 1 a and b be sub divided based on menopausal status?

Author Response

We appreciate your comments on our manuscript entitled “Targeting amino acid metabolic reprogramming via L-type amino acid transporter 1 (LAT1) for endocrine resistant breast cancer”.

We did study the association  between  LAT1 expression and menopausal status of the patients we examined in this study. 36 patients were positive for LAT1  in the premenopausal group (N = 67, 54%), and 60  in the postmenopausal group (N = 120, 50%) ( p:  0.624). We studied using  the Chi-squared test. As above, there were no significant differences of LAT1 status according to the menopausal status of the patients in this study.    In addition, we also examined the correlation between LAT1 expression and therapeutic response  to aromatase inhibitor (AI) treatment because of the previously reported  association between LAT1 expression and tamoxifen-resistance. In particular,  we examined postmenopausal  patients who received neoadjuvant hormone therapy (NAH) in the second cohort.  A high post-LAT1 expression was significantly correlated with the disease stage (P = 0.0003), pathological T stage (P = 0.0096), pathological N stage (P = 0.0365) and histological grade (P = 0.0058) of the patients. Conversely, a low post-LAT3 expression was significantly correlated with the disease stage (P = 0.0474) and the pathological T stage (P = 0.0386) of the patients.

Round 2

Reviewer 2 Report

Accept